# Comparative Study of Pine Reference Genomes Reveals Transposable Element Interconnected Gene Networks

**DOI:** 10.3390/genes11101216

**Published:** 2020-10-16

**Authors:** Angelika Voronova, Martha Rendón-Anaya, Pär Ingvarsson, Ruslan Kalendar, Dainis Ruņģis

**Affiliations:** 1Genetic Resource Centre, Latvian State Forest Research Institute “Silava”, LV2169 Salaspils, Latvia; dainis.rungis@silava.lv; 2Linnean Centre for Plant Biology, Department of Plant Biology, Swedish University of Agricultural Sciences, SE-750 07 Uppsala, Sweden; martha.rendon@slu.se (M.R.-A.); par.ingvarsson@slu.se (P.I.); 3Department of Agricultural Sciences, University of Helsinki, FI-00014 Helsinki, Finland; ruslan.kalendar@helsinki.fi

**Keywords:** pine reference genome, gene networks, gene regulation, node gene, transposable elements, retrotransposons, MITE, *Pinus taeda*, *Pinus lambertiana*, introns

## Abstract

Sequencing the giga-genomes of several pine species has enabled comparative genomic analyses of these outcrossing tree species. Previous studies have revealed the wide distribution and extraordinary diversity of transposable elements (TEs) that occupy the large intergenic spaces in conifer genomes. In this study, we analyzed the distribution of TEs in gene regions of the assembled genomes of *Pinus taeda* and *Pinus lambertiana* using high-performance computing resources. The quality of draft genomes and the genome annotation have significant consequences for the investigation of TEs and these aspects are discussed. Several TE families frequently inserted into genes or their flanks were identified in both species’ genomes. Potentially important sequence motifs were identified in TEs that could bind additional regulatory factors, promoting gene network formation with faster or enhanced transcription initiation. Node genes that contain many TEs were observed in multiple potential transposable element-associated networks. This study demonstrated the increased accumulation of TEs in the introns of stress-responsive genes of pines and suggests the possibility of rewiring them into responsive networks and sub-networks interconnected with node genes containing multiple TEs. Many such regulatory influences could lead to the adaptive environmental response clines that are characteristic of naturally spread pine populations.

## 1. Introduction

The functional role of transposable element (TE) insertions distributed throughout plant genomes is less studied and phenotypic changes are less obvious compared with protein-coding regions [1,2,3]. Transposition of TEs is linked to stress conditions and evolutionary change [3,4,5,6,7,8,9,10,11,12], but these sequences are usually controlled by the host organism [13]. Progress in transcriptome sequencing has revealed multiple types of non-coding RNAs that originate from TE sequences, nested elements, and their relicts remaining after purifying selection [3,14,15,16]. Information from various plant species and genes where TE-derived insertions are linked to phenotype alterations is accumulating [17,18,19,20,21,22,23,24]. Reported influences of these insertions include gene interruption by transposition; alteration of gene expression levels via providing additional transcription initiation signals or downregulation by methylation; exon shuffling and alternate splicing; initiation of antisense transcription; non-coding RNA production or providing target sites; providing additional poly-A signals that change transcript stability and transport from the nucleus; multiple insertions of similar TEs that could establish dynamic gene networks [10,25,26,27,28,29,30].

TEs are organized into classes (retrotransposons and DNA transposons), superfamilies (Ty1-*copia*, Ty3-*gypsy*, etc.), families, and subfamilies. Up to 20% nucleotide sequence variation is permitted between family members [31]. The most prevalent TEs in plants are long terminal repeat (LTR) retrotransposons (RLXs), which are sequences that contain direct repeats that flank the internal sequence or body of the element. LTRs may contain transcription initiation (plant type II promoters) and termination signals (polyadenylation site), polypurine tracts, integrase-binding signals, tRNA primer binding sites (PBS), and cis-acting elements [32]. Several elements with extraordinarily short LTRs have been described for angiosperms (e.g., 85 bp, *FRetro129* [33]), while the LTRs of other RLXs can be longer than 5 kb in length (*Sukkula*, [34,35], *Grande*, [36], and *Ogre*, [37]). However, these are exceptions and LTRs are typically 0.1–2 kb in length [38,39]. The internal region of autonomous RLXs contain gag- and polyprotein-coding domains that produce the proteins necessary for retrotransposition, namely, protease, reverse transcriptase, and integrase. Non-autonomous elements contain empty, partial, or disrupted polyprotein sequences and use proteins produced by autonomous elements, sometimes integrating more effectively than their autonomous partner [35,40,41,42]. Reverse transcription processes introduce mutations and can produce chimeric elements by template switching [43]. Non-homologous recombination often involves highly similar repeats and can result in deletions of portions of TEs or even neighboring genes [44]. Some genome regions contain multiple insertions of TEs into each other, forming nested repeat regions [45]. During transposition, RLXs proliferate via RNA transcript intermediates, but DNA transposons excise from their current location and migrate to a new genomic locus. Therefore, copy number proliferation is not as pronounced for DNA transposons. However, some non-autonomous DNA transposons have been found more frequently in plant gene regions, such as Miniature Inverted-repeat Transposable Elements (MITEs) [46,47,48]. Distributed TE families are broadly used as molecular markers in population genetic investigations and marker-assisted breeding [48,49,50,51,52,53,54,55,56,57].

Conifer genomes are greater than 15 Gbp in size and contain extended regions of non-coding DNA, much of which are derived from TEs. The genomes of pines represent one extreme in plants, with stable diploid genomes that are expanded by the proliferation of TEs, in contrast to frequent polyploidisation events in angiosperms [58,59,60]. TEs are rarely deleted via non-homologous recombination processes; therefore, the ratio of solo LTR to full-length elements is lower in conifers, and most TEs are represented as full-length elements [60]. The genome sequences of several gymnosperm species have been published [60,61,62,63]. However, data quality, coverage, and gene models are continually improving [64]. The loblolly pine (*Pinus taeda*) genome contains a high proportion of TEs; there are approximately 1500 families with the prevalence of LTR RLXs (42% of genome sequence [65,66]. In conifer genomes, TE sequences are highly diverged; however, some conserved families are found in *Pinaceae* and even in more distant gymnosperms [67,68]. Our previous studies on Scots pine (*Pinus sylvestris*) revealed a rapid expression of TE-containing sequences in response to stress conditions [69]. The composition of the studied RLXs was found to be specific to pine lineages, while family proportions and copy numbers showed variation between and within pine species [68]. The more frequent distribution of a particular RLX family in the pine genome does not lead to its higher expression rate, but expression levels of different RLXs were strongly correlated within individual seedlings [70]. Therefore, we hypothesized that stress-responsive genes or their surrounding regions could be enriched with particular TE families that are co-expressed with genes under unfavorable conditions. The aim of this study was to analyze genes containing TEs and the distribution of TEs in genomic sequences of genes and gene-flanking regions in the available pine reference genomes (*P. taeda* and *P. lambertiana*). We also explored the possibility of transferring this information to *P. sylvestris* genome studies. The obtained results were used to evaluate if the distribution of TEs in gene regions is random regarding different gene regions (e.g., flanks or introns), species or TE families; if genes containing similar TE families are involved in similar processes; and if the investigated TEs contain potential gene regulatory motifs. 

## 2. Materials and Methods

### 2.1. Generation of Datasets

Reference genomes for *P. taeda* v.1.01 and v.2.0 and *P. lambertiana* v.1.01 were downloaded from https://treegenesdb.org. The *PIER* v.2.0 (Pine Interspersed Element Resource) [65] database was used for TE identification. The *UPPMAX* (Uppsala Multidisciplinary Center for Advanced Computational Science) resource was initially used for analyses of genomic sequences. The Riga Technical University High Performance Computing (HPC) Centre was used for further analyses in Latvia. The publicly available PIER 2.0 database contains conifer TEs and nested repeats that were recognized by automatic genome annotation from the *P. taeda* v.1.01 genome. A large portion of the database entries contain nested repeats with 2–4 pairs of direct LTRs rather than full-length TE sequences, which could negatively influence the analyses in this study. One TE insertion within a particular gene will have hits to hundreds of different database entries containing high sequence similarity to this TE; therefore, all full-length elements were clustered with *CD-Hit* v.4.6.4 [71] utilizing a sequence identity threshold of 0.8. This resulted in 15,622 unique TE representatives from the 19,700 entries originally found in the database. This was the full-length TE database used for analyses. A further problem was the high sequence diversity of conifer TEs that contain many insertion-deletion polymorphisms. It is possible to apply strict similarity search criteria, but then two genes containing insertions of the same TE family will be not recognized as members of the same network. In this case, potentially important carriers of similar transcription signals (e.g., LTR sequences) will not be recognized. Additionally, genome scaffolds are often truncated at repeat insertion loci, leaving numerous genes with only partial TE sequences unmasked. Therefore, all direct repeats were extracted from each defined mobile element in the *PIER* database, resulting in 24,591 repeats. The new data set of extracted repeats contained sequences from 79 bp to 13,295 bp in length, with an average length of 579 bp and a median length of 417 bp. The presence of nested repeats could lead to the extraction of direct repeats of the same TE family or sequences that represent a body of nested TEs. This was verified by individual sequence inspection and all-to-all alignment. To reduce such errors, extracted TE-derived repeats were *CD-Hit* clustered, resulting in 9659 entries. Sequences of only 0.1–2 kb in length were used for further analyses, resulting in a final database of 9107 entries (5.7% reduction). This was the TE-derived repeat database.

### 2.2. Extraction of Gene Introns and Flanking Regions

*Bedtools* v.2.27.1 [72] were used to extract sequences of all genes containing introns from the reference genomes. Full-length and TE-derived repeat similarity searches within the extracted gene regions for each reference genome were performed with basic local alignment search tool (BLAST) *BLASTn* v.2.2.26 [73]. *Samtools* [74] were used for matching gene sequence extraction. *BEDOPS* v.2.4.35 [75] used for extraction of gene-flanking regions, with 5-kb sequences from the 5′ and 3′ flanking regions of each gene extracted into separate databases. Flanking sequences were further divided into 1-kb fragments, resulting in 10 databases. If any of 1-kb region included a scaffold end, the shorter remaining sequence was included. Each flanking region was referenced to its gene ID and position. Sequences were compared using *BLAST* with the following parameters: percent identity ≥80%, alignment length ≥100 bp, Query Coverage Per High-scoring Segment Pair (HSP) ≥90%. For *P. lambertiana* analyses, the Query Coverage Per HSP parameter was lowered to ≥80% as the previous parameter set produced no hits. A *t*-test was calculated for the evaluation of TE enrichment with significance to flanking regions in the vicinity of genes compared with other regions. The standard deviation of the distribution of differences between independent sample means was estimated for each TE-derived repeat (for the equal variance *t*-test). Enrichment significance of *p* = 0.001 was considered if *t* was greater than 5.04 (df = 8) and *p* = 0.05 if *t* > 2.31 (df = 8).

To overcome inconsistencies found in genome files, *P. taeda* v.2.0 true genomic coordinates were evaluated for 15,534 transcripts annotated with any Gene Ontology term (from 36,730 entries annotated as genes). Correct genomic coordinates of transcripts were identified by running a computationally intense local NCBI short-blast algorithm with parameters of sequence identity >98% and 100% query coverage per HSP. The resulting genomic coordinates were further sorted and corrected as follows: scaffolds containing identical hits and gene coordinates were deleted, with genes larger than 20 kb manually verified and repetitions of partial exon structures or gene repeats on one scaffold deleted, leaving only one conventional structure with all exons present. Further gene transcripts were screened against the *PIER* database, and genes matching TEs with more than 50% query coverage and more than 80% sequence identity were filtered out. All extractions and analyses were repeated with the new gene set. After all steps, genes containing two or more types of repeats (filling less than 50% of template) were still present in the dataset and were later filtered from the results. Gene genomic sequences from the transcription start to termination site were extracted from the reference genomes into separate databases.

### 2.3. Analysis of LTR Structure and Transcription Factor Binding Sites (TFBS)

The database of LTR representatives was used for analysis of flanking and intron regions, which was generated by computational selection of any repeated region found in one RLX sequence defined by automated predictions. Then, 5-kb gene-flanking regions containing hypothetical distributed LTRs were extracted and full-length TE structures were identified if possible. Further, verified TE structures were repeatedly queried among all gene regions and the related information was updated. Hits with ≥1 kb query coverage and more than 80% nucleotide identity to full-length TEs were considered strong candidates. Consensus sequences were evaluated after multiple alignment of extracted TE sequences. Characteristic features of TEs were identified using an NCBI Conserved Domain search, the *REPFIND* repeat prediction tool [76], and *Repbase* [77]. *Softberry* tools (ScanWM-PL, TSSPlant, NSITE-PL, POLYAH, http://www.softberry.com; [78]), and *PLACE* v.30.0 [79] were used for the identification of TFBS, plant promoters, and poly-A sites in LTRs that were found frequently near exons. *miRBase* [80] and the *RNAfold* web server [81] were used for microRNA prediction.

### 2.4. Gene Networking and Gene Ontology Analysis

The Gene Ontology (GO) classification file was obtained from the Gene Ontology Consortium (http://www.geneontology.org/), and the *P. taeda* v.1.01 gene functional annotation was downloaded from *PLAZA* [82]. Gene functional annotation for *P. taeda* v.2.0 was generated by considering gene transcript homology to *P. taeda* v.1.01, which enabled comparison between differing numberings of genome versions. Fewer than 50% of the genes were categorized to any GO term for *P. taeda* v.2.0 (15,534 of 36,730). For *P. lambertiana* v.1.01, a functional annotation file was downloaded from the *Treegenes* data repository (https://treegenesdb.org/). In total, 8943 genes from 13,637 were assigned with any GO term for *P. lambertiana* v.1.01. Homology of the evaluated gene set in each network was tested by sequence comparison of transcripts (whole set) between two species (*e*-value < 0.001).

Gene networking analysis was performed only to assess potential functional significance of the evaluated gene groups. Each gene network was established by the presence of high-quality matches to similar TEs (members of one TE family), present in a particular location (5′ and 3′ 0–2 kb gene flanks or introns). High-quality sequence matches (>1 kb, >80% sequence identity to TEs; for short MITE elements > 200 bp matches; *e*-value < 0.01) were considered in the final networking analysis. *BINGO* v. 3.0.3 [83] was used for gene networking analysis using the custom annotation available, with *Cytoscape* v.3.3.0 [84] used for gene network visualization. Gene networks were formed from the annotated genes, as genes without annotation were automatically excluded by the software. Network edges represent connections of GO terms, but the node size depends on the gene count categorized to a particular term. A hypergeometric test with Bonferroni correction implemented in *BINGO* was used for the GO term overrepresentation test in one particular network compared to custom GO annotation of all pine genes. Comparative networks between two species or different TE locations (flanking regions or introns) were constructed using *DyNet* [85]. Further, for updating gene annotations, network-associated gene transcripts were extracted and additionally annotated, using a blastx search against the NCBI reference protein database with *CLC Genomics Workbench* Blast2GO analysis workflow tools. The *GenAlEx* 6.05 software package [86] was used for analysis of TE insertion patterns. Identified protein interactions were analyzed using *STRING* v.11 [87]. A graphic overview of the analysis workflow is provided in Appendix A.

## 3. Results

### 3.1. Quality of Assembled Genomes and TE Assay

The quality of reference genomes and the repeat database used (full-length vs. TE-derived repeats databases, see methods) play a major role when analyzing the presence of TEs in gene regions. Analysis of the first version of the *P. taeda* genome with the full-length TE element database identified more genes containing TE sequences in introns (the top-ranked TE had unique matches to 200 genes) than in the *P. taeda* v.2.0 genome assembly (the top-ranked TE had matches to only 39 genes). A similar situation was observed when analyzing the 5-kb flanking regions using the full-length TE database. However, when the TE-derived repeat database was used for the analyses, the number of genes with repeats in the introns or flanking regions was similar in both *P. taeda* genome versions. The main reason for these differences may be because the full-length TE elements identified from v.1.01 did not align over their entire length with the v.2.0 genomic sequence assembled from longer reads. Additionally, observation of particular gene introns with similarity to interspersed repeats revealed that incomplete TE sequences in v.2.0 were frequently present with masked sequences nearby. This suggests persistent problems with TE-containing read assembly and indicates the need for validation of observed matches to confirm the presence of full-length or partial TEs. Due to the diversity and repetitive nature of conifer TEs, the differentiation of full-length elements from partial, chimeric, or nested copies is often not possible without resequencing. The presence of any part of a TE could be indicative of the occurred insertion, and the partial or full-length structure could be further confirmed in additional experiments. Therefore, utilizing shorter repeats as representatives was more suitable at this point for evaluation of prevalent TEs in or near genes. For *P. lambertiana*, only one genome version is currently available, and using the full-length TE database, the most frequent families were found in the vicinity of 272 genes, which was comparable only with results from the *P. taeda* v.1.01 genome. However, utilizing the short TE-derived repeat database, approximately 500 matching genes were found for the most frequent TE family in both *P. taeda* v.1.01 and v.2.0, and 532 genes were identified in the *P. lambertiana* genome, which was comparable across all versions. Evaluated amounts of TEs are in accordance with whole-genome TE content data, where the larger genome size of the pine subgenus *Strobus* (*P. lambertiana*) correlates with a higher gene and repetitive element content [62].

Previously annotated pine RLXs were used to assess the quality of genome assembly in gene regions. The ratio of LTR to internal sequences of RLXs should reach a value of two if RLXs are present as full-length elements. However, incomplete sequences of the ancient and widely distributed *IFG* family were found in gene introns of *P. lambertiana*, with more internal sequences present than LTRs (Appendix A). In contrast, in the *P. taeda* v.2.0 genome, solo LTRs of *IFG* prevailed. More rapid degradation of LTR sequences is common for RLXs in plant genomes; however, according to previous studies, this process is relatively slow in conifers. Moderately distributed TE families, such as *Pinewoods* and *Appalachian*, were represented in both species’ genomes as full-length sequences (Appendix A). However, the low-copy-number RLX *Angelina* was overrepresented by solo LTRs in *P. taeda* introns, as only one complete internal sequence and 13 LTRs were found at a 90% similarity level. Generally, at 80% sequence similarity, more solo LTRs were identified, which could be a result of genome assembly from short-reads, fragmentation of RLX, structural differences, or degradation of some sequences in gene introns. These results indicate that the structural inconsistencies are present in the current assemblies, not only across the entire genome, but also in the functional regions of genes. 

### 3.2. Topology of TEs in Gene Non-Coding Regions

An overview and comparison of the number of extracted gene-flanking regions and total number of hits to TE-derived repeats in 5′ and 3′ gene-flanking regions between the two pine species are presented in Table 1. The ratio of hit number to the number of extracted flanking regions was similar for all regions in the *P. taeda* v.2.0 genome (0.1–0.11, except for the 0–1 kb region with a ratio of 0.16–0.18). After in-depth analysis of the *P. taeda* v.2.0 genome, global errors in gene coordinates were revealed such that many protein-coding gene transcript coordinates were mapped to TE genomic coordinates without any sequence similarity. This increased the number of LTRs found in the 0–1 kb flanks of false genes. After creating our own genome annotation (see methods), similar tendencies were revealed in the TE distribution in all studied genomes, namely, an increase in hits to TE-derived repeats with increasing distance from genes.

The *P. taeda* v.2.0 genome currently has the best scaffold quality regarding TE location. To analyze the distribution of TEs relative to distance from genes, the 5-kb flanking regions of filtered *P. taeda* v.2.0 genes were divided into 1-kb regions (0–1; 1–2; 2–3; 3–4, and 4–5 kb from genes); the distance and the position (5′ or 3′) relative to each gene were therefore retained. For each repeat, t-statistics were calculated comparing average unique hit numbers to the gene-flanking regions. In total, 135 different TE-derived repeats were identified in gene-flanking regions. Significantly increased occupancy in the 0–1 kb gene-flanking regions was determined for nine (*p* = 0.001) and seven TEs (*p* = 0.05) compared to the 2–5 kb flanking regions (Appendix A). Of these enriched TE families, occupancies of two TEs (*p* = 0.001) was also significantly increased in the 0–2 kb flanking regions. The number of genes where a particular TE was enriched in the 5′ and 3′ 0–1 kb flanks ranged from 27 to 48 genes. False genes were identified and evaluated in subsequent detailed sequence analyses. These false gene sequences consisted of parts of several different TEs that were not present in databases and therefore could not be filtered out in previous stages. Therefore, fewer protein-coding genes actually contained TEs in their vicinity than initially identified, and all gene groups were repeatedly screened against the NCBI database using a blastx search to confirm that they were protein coding. 

Analysis of enriched repeats in gene-flanking regions allowed identification of a MITE element in both species’ genomes, which we have named *Plater* and describe in further detail below. In addition, we identified evidence of the distribution of a *Copia* RLX *RYX6* LTR family in the 0–2 kb flanks of *P. taeda* genes. This element had 91% similarity to the *PpRT6* partial RNaseH-like gene (EF102091) previously reported for *P. pinaster* [88]. The full-length element (3367 bp) was present in only one gene flank while others contained solo LTRs; most sequences between LTRs were masked. The identified potential LTR was rich in AT-containing motifs (AT-1, ARR1AT, CAATBOX1, MYB1AT, CIACADIANLELHC, MYCCONSENSUSAT, ROOTMOTIFTAPOX1, P1BS, “TATCCA” element and others). The extracted 416-bp LTRs contained conventional 5′-TG-CA-3′dinucleotides, TATA box (S000203), and a polypurine tract. The absence of full-length coverage of *RYX6* due to masked regions prevented determination of a consensus sequence and verification of the results, indicating that sequence scaffolding problems persist in the case of longer repetitive elements. Another enriched repeat identified belonged to the 5767-bp RLX *Copia-2602* with 160-bp LTRs. After filtering of false-positives, the hit count to the newly identified RLX increased with distance from the gene. The relatively short LTR of this RLX contains three TATA box motifs and three CAAT boxes on the negative strand; therefore, insertion of this element could promote antisense transcription. Only two *P. lambertiana* genes possessed a *Copia-2602* insertion in 0–1 kb flanks; these were a probable histone H2B.3 gene and a Piwi-like domain containing argonaute family protein-coding gene. In the *P. taeda* 2.0 genome, two gene flanks contained hits to *Copia-2602* (serine/threonine protein kinase PEPKR2 and carnitine transporter 4). Insertions of the *Copia-2602* LTRs into 3′ flanks of a TMV resistance protein N-like gene and one unknown gene were found. *Copia-2602* was also identified in the 1-2 kb flanks of flavonoid 3′,5′-hydroxylase 2, probable disease resistance protein At4g33300, chitinase 2-like, and cytochrome P450 (CYP736A12) genes.

In conclusion, in the *P. taeda* and *P. lambertiana* genomes, all gene sets contained almost no highly frequent TE insertions in the 0–1 kb gene-flanking regions (with the exception of MITE). Therefore, nearby gene regions were characterized with moderate TE distribution and diverse TE insertions. Furthermore, the number of most TE-derived repeats gradually increased with distance from genes, suggesting a slight elimination of TEs from the gene vicinity. Most TE insertions accumulated in the gene introns (Figure 1). The patterns of the most widely distributed TEs in the gene introns of both species were compared (Figure 2, Appendix A). Far more short TE-derived repeats were evaluated in the gene introns of both species (Appendix A). Interspersed repeats that were identified as associated with genes in both the *P. taeda* and *P. lambertiana* genomes were manually reviewed to verify their full-length structure, and TEs with high-quality hits (>1 kb in length) were considered for further networking analyses (Figure 2).

Gene transcripts containing identical TEs in their genomic sequences were compared, but in most cases no homologous transcripts between *P. taeda* and *P. lambertiana* were found. TEs distributed in the gene non-coding regions of both species were analyzed in more detail and only the structurally most well defined and informative are described separately in the next sections. Evidence of the presence of other TEs in gene flanks and introns was also found; however, the fragmentation of the current genome assemblies does not allow these data to be confirmed conclusively.

### 3.3. Gene Ontology Analysis

To obtain an overview of the functions of genes with similar TE insertions in introns or flanks, Gene Ontology (GO) analysis was performed for the extracted gene groups. However, approximately 50% of all pine genes were not annotated to any GO category and therefore these genes were automatically excluded from the networking analysis. 

For flanking regions, GO analysis was performed for genes with a statistically higher frequency of TE-derived repeats (Appendix A). Some members of the identified gene networks contained TE insertions in their vicinity with the highest t-significance of enrichment, but had few associated GO terms and were poorly annotated. TE-containing transcripts were annotated with GO terms such as DNA integration, RNA-dependent DNA biosynthesis, endopeptidase activity, or nucleic acid binding. Some gene groups contained only a few annotated genes, and their biological significance could not be established with certainty.

A higher diversity of TEs were identified within gene introns. Gene networks were built for genes having similar TE insertions in their introns without taking into account the exact location of TEs regarding intron number and position (Appendix A). GO categories from biological processes and molecular function were determined, and overall hits to unique genes and the GO-annotated gene count in category were indicated. In the *DyNet* comparative trees, common GO categories shared between the two species are indicated in white (Appendix A). Genes from the two species containing identical TEs were categorized into similar parental GO categories. However, no homologous genes were revealed in most of the analyzed networks between the two pine species. This demonstrates that gene regions of two distantly related pine species contain insertions of TEs in different genes involved in similar processes. The identified gene networks were often associated with defense and regulative responses, such as oxidation-reduction processes, transmembrane receptor biosynthesis, metal ion binding, hormone metabolic processes, and carbohydrate metabolic processes (Appendix A). 

### 3.4. MITE Plater Distribution in Gene Vicinity and Introns

A TE-derived repeat *PtRXX_3321* was significantly enriched in the *P. taeda* v.1.01 gene flanks (0–2 kb) and was also frequently found in gene introns (Appendix A). In-depth analysis of the sequences obtained from the *P. taeda* v.2.0 genome revealed a MITE element, subsequently named *Plater*, that was 259 bp in length and contained 24-bp terminal inverted repeats (TIRs). The stem sequence was flanked with 40-bp direct repeats which were identified in the first intron of the probable pectinesterase/pectinesterase inhibitor gene. A consensus sequence of the *MITE* transposon *Plater* was built for each species genome. Compared with *P. taeda*, the *P. lambertiana Plater MITE* contained a 10-bp insertion in the 3′ stem sequence and was 265 bp in length with 28-bp TIRs. The *P. taeda Plater* element (sharing more than 80% sequence similarity) was found in 74 gene introns, of which 58 unique gene introns contained *Plater MITE* insertions with 99–100% nucleotide identity (Figure 3). The *P. lambertiana* high-quality gene set contained 87 genes with *Plater MITE* within introns; however, the diversity of these sequences was higher (94–96% nucleotide similarity with the species consensus sequence). *Plater MITE* insertions were identified in the vicinity of 191 *P. taeda* genes and of 65 *P. lambertiana* genes (Figure 3A). Gene introns containing several *Plater MITE* insertions were analyzed. An unannotated *P. taeda* gene with a conserved phosphoglucosamine mutase family protein domain contained a maximum of seven *Plater* MITEs (Table 2). 

GO categories common to both species included the regulation of gene expression, response to stress, and transmembrane transport. For example, genes carrying *Plater MITE* in introns share GO categories such as glycotransferase activity, where the *P. lambertiana* gene had 1.3-beta-D-glucan synthetase activity, but the *P. taeda* gene had xyloglycotransferase activity (Appendix A). However, gene nucleotide sequence comparisons revealed no homologous genes between the evaluated *P. taeda* and *P. lambertiana* genes containing similar *Plater* insertions in genomic sequences. This, as well as the revealed species-specific structural differences, suggests that *Plater MITE* was found in a common ancestor, but expansion of this element occurred after the separation of species. A comparison of genes containing *Plater MITE* within introns with genes having identical MITE in their flanks did not reveal any common genes in either species, indicating that *Plater* was inserted into either introns or into the flanks of certain genes, but never in both sites of presumably transcriptionally active genes (Figure 3). Therefore, insertion of *Plater MITE* could not be explained only by random transposition into transcriptionally active chromatin.

Better-annotated *P. taeda* genes containing *Plater MITE* in introns or in flanks were compared (Appendix A), revealing that genes containing *Plater* insertions in their flanks were involved in the regulation of developmental processes, such as regulation of cell division, pollination, negative regulation of macromolecule biosynthetic processes, nuclear division, and methylation. Genes containing *Plater* insertions in their introns were involved in innate immune response, positive regulation of defense responses (jasmonic acid-related responses), pigment metabolic processes, plastid organization, maturation of ribosomal proteins, potassium ion transmembrane transport, and proline biosynthetic processes (Figure 3A,B). Both networks contained GO terms related to ion homeostasis, glucan metabolic processes, proteolysis, regulation of gene expression, post-embryonic development, and oxidation–reduction processes (Appendix A). 

*Plater MITE* elements from *P. lambertiana* and *P. taeda* contain one TATABOX on the positive strand, 7–10 ARR1-binding elements, 2–4 CAAT boxes, 4–5 DOF TFBS, 3 GT-1 binding sites; these are all important regulative motifs found in the promoter regions of plant genes. The *P. lambertiana Plater* contained a 10-bp insertion (AGAGAAATTA) that disrupted a site (TTTGACC) identical to several WRKY TFBS, but gained a site identical to co-dependent regulatory elements responsible for pollen-specific activation (Figure 3 C). Differences in predicted TFBS presence in *Plater* could explain the depletion of this TE in the *P. lambertiana* 0–1 kb gene flanks and enhanced distribution in gene introns. 

### 3.5. DNA Transposon Irbe Forms a Stress-Responsive Gene Network

An 820-bp TE-derived repeat, which further facilitated the identification of the DNA TE subsequently named *Irbe*, was initially found as moderately distributed within *P. taeda* v.1.01 gene introns, and many genes contained extended GO annotations belonging to defense responses (Appendix A). One of the identified genes was Nonexpresser of Pathogenesis-related proteins-1 (*NPR1*), which is involved in plant systemic acquired resistance and the salicylic acid-mediated signaling pathway. *NPR1* contained several TE-derived repeats in the second intron and all repeats were tested with additional searches, but only *Irbe*-related repeat was distributed and found in an additional 200 gene introns from the non-filtered *P. taeda* v.2.0 genes and formed a network of stress-responsive genes (Figure 4). In the entire *P. taeda* v.2.0 genome, only 251 copies of the *Irbe* DNA TE were found, indicating preferential distribution of this TE in genes. Other identified genes included a histone-binding PHD1 finger protein ALFIN-like 4 coding gene, a COPII-coated ER to Golgi transport vehicle SNARE-like 13 gene, eukaryotic translation initiation complex 2B, ribosome biogenesis protein RPF1, and other important genes (Appendix A). Only a partial unmasked sequence of the repeat was present in the introns and therefore a full-length insertion was not confirmed for all genes.

A conserved domain search of the *Irbe* TE revealed transposase-like protein (pfam05699), hAT family C-terminal dimerization region (pfam05699), and BED zinc finger domain (pfam02892), indicating that this element belongs to the DNA TE class. The FindMiRNA tool (http://www.softberry.com/) predicted seven probable pre-microRNAs in the *Irbe* TE with free energies ranging from −53.75 to −45.82 kcal/mol. A search of MiRBase revealed homology to the mature microRNA sly-miR9472-3p from a drought-tolerant tomato line [89,90]. The *P. lambertiana* gene introns contained more hits to the newly isolated *Irbe* element; 143 genes contained hits longer than 1 kb with sequence identity >82%. *Irbe* consensus sequences from both species were compared, but no species-specific structural polymorphisms were identified, suggesting ancient transposition events and probable distribution in genes of other pine species. Gene transcripts from *P. taeda* and *P. lambertiana* containing the *Irbe* TE were compared and one homologous gene was identified with more than 95% sequence identity and 100% query coverage. This gene was annotated as 26S proteasome non-ATPase regulatory subunit 4. The *Irbe* DNA TE was not identified in the 0–1 kb gene flanks of both species.

### 3.6. Distribution of the Widespread IFG Gypsy RLX

The *IFG* RLX is a remarkable TE family, as it is highly distributed in conifer genomes and is far more ancient than other RLXs, but sequence homology is still maintained [67,68]. High-quality matches of *IFG* sequences to gene introns were considered in gene network analyses, resulting in 99 genes from the filtered *P. taeda* v.2.0 dataset and 317 genes from high quality (HQ) genes in *P. lambertiana* (Appendix A). The following three homologous protein kinase genes with *IFG* insertions were identified: plastidial pyruvate kinase coding gene, PTI1-like tyrosine protein kinase gene, and putative receptor-like protein kinase gene. The *P. lambertiana* tyrosine protein kinase had an 85,239-bp second intron with one single *IFG* insertion (internal part with 3′ attached one matching LTR), and 7987 bp in total from intron II was masked. In the *P. taeda* v.2.0 tyrosine protein kinase gene, *IFG* was inserted into the first intron, which is 106,013 bp long, and only 1639 bp was masked. Both protein kinase genes shared an identical exon-intron structure and 97% cDNA similarity. While *IFG* LTRs were 82% similar, the *IFG* body was interrupted by sequence masking. A receptor-like protein kinase gene contained exon duplication events in both species, with both introns containing one full-length *IFG* insertion on the minus strand. These insertions were 87% similar in the *IFG* body and 79–83% similar in their LTRs. The *P. taeda* v.2.0 gene contained an additional *IFG* sequence with one attached LTR on the positive strand, two *IFGs* on the positive strand, and four nested LTRs with surrounding masked regions. The *P. lambertiana* receptor-like protein kinase gene contained one full-length *IFG* and one single LTR on the positive strand.

*IFG* insertions were also found in gene flanks (Appendix A), but RLX frequency tended to increase with distance from genes in both genomes. The *IFG* LTR insertion was in close vicinity to 14 filtered *P. taeda* genes (0–1 kb) in the 5′ flanks. Genes encoding sugar transport protein 7 contained only the *IFG* body in the 5′ 0–1 kb flank; 12 genes’ 3′gene flanks contained *IFG* insertions (Appendix A), but only two annotated genes contained IFG LTRs (abietadienol/abietadienal oxidase and protein DMR6-LIKE OXYGENASE 2-like). Similarly, in *P. lambertiana*, 18 genes contained *IFG* insertions in close vicinity to genes (Appendix A). No genes were found that contained insertions of *IFG* in both the introns and flanking regions (5′and 3′ 0–2 kb) in the *P. lambertiana* or *P. taeda* v.2.0 genomes.

### 3.7. RLX Daugava Resides in Gene Introns and Flanks

A newly identified RLX *Daugava* was found frequently distributed within gene introns in the *P. lambertiana* and *P. taeda* genomes. The RLX isolated from a BAC clone was 6884 bp long with two large 1386 bp LTRs, with conventional TG and CA sites. The RLX internal region was weakly similar at the protein level to the Retrovirus-related Pol polyprotein from tobacco *tnt1* (39% identity, 59% positives). The PPT (AAGAGGGAG) site was identical for elements from *P. taeda* and *P. lambertiana*. Some inserts in other locations probably have shorter LTRs, but due to multiple masked regions, it was not possible to extract more copies for the consensus structure. The *P. lambertiana* photosystem II stability/assembly factor HCF136-coding gene contained a *Daugava* RLX in the first intron in the positive orientation; the LTR was similar to an RLX from *P. taeda* with 34% query coverage and 82% sequence identity, and the RLX body was 89% similar with 62% coverage. However, some parts of this RLX were masked. The *P. taeda Daugava* LTR contained the following two AG-rich tracts: (AGNN)_3_(AG)_3_(NNAG)_2_ and (AGNN)_2_(AGN)_4_. The *P. lambertiana Daugava* LTR also contained polypurine-rich motifs 25 bp apart: AA(AGG)_2_A_3_(AGG)_2_GA_3_AGG and GAG(AGG)_3_AGA(AG)_3_. The *P. taeda* filtered gene set contained 80 strong hits to genes with *Daugava* RLX in the introns (Appendix A). The *P. lambertiana* high-quality gene set contained 688 genes with similar hit parameters, 243 genes were not annotated, and 67 were uninformative. For the network analyses, we considered only >2 kb hits to *Daugava* RLX with strong hits to LTRs; 116 genes carrying such a combination of hits were found in *P. lambertiana* (Appendix A). There were 19 *P. lambertiana* high-quality genes containing the *Daugava*-related insertion in their flanks (Appendix A). Only the following two *P. taeda* genes contained *Daugava* repeats in 0–1 kb flanks: cytochrome P45078A7-like and secreted RxLR effector protein 161-like.

High TE diversity within gene introns was observed in both pine species, and the presence of intronic TE insertions could form gene networks with similar expression and response patterns. If each interspersed repeat introduced additional gene regulation signals in gene introns, then genes containing a single interspersed repeat might show specificity regarding their function or expression pattern. To test this assumption, genes that contained an insertion of the *Daugava* RLX family (and a lack of other TE insertions) were analyzed (Appendix A). Five of 19 *P. lambertiana* proteins were found to have homologs in the *A. thaliana* genome based on STRING software and were connected to mitotic spindle assembly checkpoint (co-expressed, found interacting and mentioned together in other publications).

The evaluated *Daugava* RLX network genes were further analyzed regarding combinations of all identified intronic TE insertions to determine if genes with specific TE combinations had similar functions. The presence of unique, short TE-derived repeats (95 loci) and unique >1 kb hits (22 loci) were used to form TE insertion patterns present in introns (Appendix A). The presence of both *Daugava* RLX and *Irbe* DNA TE was found within introns of seven *P. lambertiana* genes (Pattern type I, Appendix A). Products of these genes are involved in protein folding in ER (oxidation), positive regulation of RNA export from the nucleus, protein heterodimerization, and production of stress responsive protein. Several genes with the TE pattern type P (a combination of three RLX) are involved in pH regulation in Golgi, tethering of vesicles to Golgi membranes, nuclear protein import, and intracellular protein transport. TE pattern type Y (4 RLX) was found in the following two genes: insulinase (involved in protein targeting to mitochondrion) and histone deacetylase 15 (tag for epigenetic repression). One specific intron pattern (type C) was revealed for three *P. lambertiana* genes, which are involved in pre-mRNA maturation and splicing, and are co-expressed (score 0.168, according to the STRING database). While these analyses provide some initial clues about the formation of gene networks characterized by specific patterns of insertions of multiple TEs, further investigations are required using improved genomic sequences, gene annotations, and expression data for pine species.

The GC content was calculated for pine TEs (Appendix A). The TE-associated GC content for each gene involved in the *Daugava* RLX network was evaluated (Appendix A). However, the GC content for full-length introns could be biased due to masked TE parts in the gene introns and flanks. Therefore, patterns of identified TEs were considered, and the GC content of each full-length TE was counted for each gene intron. The overall average GC content for introns was 39% for *P. taeda* and 41% for *P. lambertiana*, which were even lower if only short hits to LTRs were considered (27% and 36%, respectively). This could be explained by the lower GC content of LTRs of some RLXs distributed in the gene introns. The mean GC content of the gene transcripts (involved in the *Daugava* RLX network) was 44% for *P. lambertiana* and *P. taeda*, which was higher than any average estimate for introns and published estimates for whole BAC clones and whole genome sequences of gymnosperms (38%, [91,92]).

### 3.8. Genes Appearing in Many TE-Associated Networks

Genes with broad GO annotations were frequently found across many predicted TE networks, which could be related to the number of different TEs found within introns of both species. Therefore, genes containing multiple TEs in their introns were isolated and analyzed. In the *P. taeda* v.2.0 genome, 75 genes were identified containing more than eight different unique TE-derived repeats in their introns, with the two-pore potassium channel coding gene containing the maximum of 65 different repeats. According to the PLAZA Gymnosperm database, *P. taeda* contains 14 members of the two-pore potassium channel gene family, which contains 659 co-occurring terms, indicating the involvement of these gene products in many plant-cell processes [93]. Other repeat-rich genes identified in *P. taeda* v.2.0 were chloroplastic/amyloplastic 1,4-alpha-glucan-branching enzyme coding gene, GTP binding protein Der, S-formylglutathione hydrolase, cytochrome P450, B3 domain-containing transcription repressor VAL2, serine/threonine protein kinase GRIK1, ribosome production factor 1, jasmonic acid-amido synthetase JAR1, COP9 signalosome complex subunit, chaperone protein ClpB3, and others (Appendix A).

In the *P. lambertiana* HQ gene set, 59 genes containing 18 to 34 different unique TE-derived repeats were analyzed. The gene that contained the maximum of 34 interspersed repeats in introns was not annotated, but could be characterized as having SpoT (COG0317i) and ubiquitin-like fold (cl28922) conserved domains. A search of the Uniprot database (The UniProt Consortium, 2019) with this gene identified an HD domain-containing protein with 63% identity and 75% positives (*e*-value 0.0) that is involved in guanosine tetraphosphate metabolic processes (GO:0015969) with 107 co-occurring terms. A gene containing the second largest number of TEs in *P. lambertiana* was also an unknown protein containing the AMN1 domain (cl2816) and F-box domain (pfam12937), annotated only with the parent term “protein binding”. Other genes with a high amount of TEs in *P. lambertiana* were DNA repair helicase XPD, mitochondrial substrate carrier family protein, translation initiation factor elF-2B, syntaxin-81, ADB-ribosylation factor, acyl-CoA dehydrogenase IBR3, and others (Appendix A). 

Interestingly, eight homologous genes were identified in both *P. taeda* and *P. lambertiana* (Table 3), from which the following two genes that share significant similarity based on their transcripts were found: plastidial pyruvate kinase 2 and phospholipid:diacylglycerol acyltransferase genes. Pyruvate kinase is involved in carbohydrate degradation and is associated with 26 GO terms. *P. taeda* and *P. lambertiana* introns contain seven common TE-derived repeats in pyruvate kinase homologous genes; O-acyltransferase activity (GO:0008374) contains 329 co-occurring terms, and 12 TEs were similar in gene homologs. SrmB conserved domains were present in ATP-dependent RNA helicases, but these gene transcripts were not identical. ATP-dependent helicase activity (GO:0003724) contained 929 co-occurring terms. A similar situation was found with proteins containing conserved domains of chromosome segregation protein SMC (cl37069) from nuclear pore complex protein NUP62, WD40 domain from protein WRAP73, RAE1 and actin-related proteins (found in a number of eukaryotic proteins that cover a wide variety of functions), alpha-tubulin suppressor (ATS1) domain-containing proteins (cl34932), and mitochondrial carrier protein domain (pfam14560, involved in localization, transmembrane transport, amide biosynthetic process and translation).

## 4. Discussion

Automated TE detection relies on several strategies, such as searches for sequences with homology to known elements, *de novo* evaluation of repeated elements, analysis of the presence of specific structural features, and combined techniques [94]. In the genome assembly process, repeated and highly similar sequences such as TEs are source of errors and gaps [95,96]. Other properties of plant genomes, such as multiple gene families, pseudogenes, and chromosomal and plastid genome duplications, further complicate the process of genome assembly and annotation [97,98]. Typically, gymnosperms are characterized by large genomes with proliferated TEs, high levels of heterozygosity, a constant chromosome number, and very rare polyploidy events [58,99]. Several conifer genomes have been sequenced using short-read assembly methods [60,62,66,100]; the *P. taeda* v.2.0 genome currently has the highest quality conifer genome regarding scaffold length. Version 2.0 was improved by merging small contigs; while the first version of the *P. taeda* genome contained 16.5 million contigs, the second version contains only 2.9 million [63]. However, the average *PacBio* read length used to reconstruct genome v.2.0 was 9665 bp with 12x coverage, which is shorter than many RLXs. The mean TE length in the PIER database is 6273 bp (median 5383 bp) but 3046 TEs are longer than 10 kb with the availability of better quality or longer-read genome assemblies, TE identification should be repeated *de novo*. Genome annotation of conifers is an ongoing process. Indeed, our study revealed errors in annotation files and datasets, where gene IDs of transcripts did not coincide with genomic sequences bearing identical gene IDs, and genes had noticeably varying intron lengths in different genome versions. Some additional errors involve automated annotation, nested repeat annotation, pools of identical reverse transcriptase domains defined as different genes, and intense masking of ambiguous regions, which are not discussed sufficiently in associated publications. Consequently, comparison of genomes assembled with different approaches and of varying quality should be performed with caution to avoid errors when interpreting the results. Additionally, widespread gene capture by TEs has often been described for large plant genomes [42,101,102,103] and this phenomenon could introduce additional errors in short-read genome assemblies [103]. Considering the high copy number and the structural and functional differences from protein-coding genes, it is advisable to annotate TE-associated sequences in separate data sets even though many TE families bear open reading frames. Despite the fact that many conifer TEs only show weak similarity to annotated TE domains, complexity was effectively reduced in the *P. lambertiana* high-quality gene set [64]. Evaluation of the only prevalent MITE element in the flanking regions of initial genome versions demonstrates the importance of read length in assembly and the association of TEs to particular genome locations.

Diverse conifer RLXs sharing high partial sequence similarities may not be classified in one family by full-length alignments used in bioinformatic studies, resulting in inflated family counts [66]. Due to the high diversity of conifer TE sequences and patchy distribution within short read assemblies, we found that identification of short repeated regions could overcome these problems and more efficiently identify prevalent TE families. The use of clustered TE-derived repeats extracted from automatically predicted nested regions allowed not only for the identification of RLXs but also internal regions of other TEs. The use of TE-derived repeats allowed simple statistical tests of distribution relative to distance from genes. RLXs are the most widely distributed TE class in plants and gymnosperm genomes [32,60,66,104]. LTRs contain important regulatory signals that can influence gene expression even if the body of the element is deleted [22,25,105]. The PIER v.2. database entries contain nested TEs that are derived from automated predictions, including repeated internal regions of different TEs. Chimeric TEs that could evolve from two RLXs by template switching [43] and display signals of independent transposition have been identified in other plant genomes, e.g., *Veju* [106] and *BARE-2* [107]. TE structures with several LTRs are found in plant genomes, for example, 13 *Cassandra* elements in pear contain three LTRs [108]. Considering the large conifer genome sizes, such chimeric structures may also be found in conifer genomes. Therefore, each high-frequency repeat should be verified, and the true full-length structure and target site duplications should be identified. Unfortunately, the pine genomes contain many low-copy-number TEs, and if all copies are masked, full-length structures or transposition sites cannot be resolved. Additionally, similarity to known proteins and GO annotations are available for only approximately half of all pine genes, but species-specific genes could be more important in the adaptation of species. Therefore, the data evaluated and the results obtained from this study could be expanded with improvement of genome quality, annotation, or other associated information.

In the current study, we identified several TE families that unite many pine stress-responsive genes and contain potentially important gene regulatory signals. The *Irbe* DNA TE could provide microRNA target sites or produce microRNA, or both. However, further analysis is required to determine if these sequences represent microRNA precursors or, alternatively target sites [109]. This DNA TE insertion was found in the second intron of the *P. taeda* NPR1 gene, a key regulator of systemic acquired resistance [110], the PSMD4 gene coding for the 26S proteasome (involved in the ATP-dependent degradation of ubiquitinated proteins), ubiquitin thioesterase OTU1 (plays an important regulatory role at the level of protein turnover by preventing degradation), S-formylglutathione hydrolase (detoxifies formaldehyde), a PHD finger protein ALFIN-LIKE 4 (a histone-binding component that recognizes H3K4me3), and other important genes. Based on the presence of *Irbe*-related repeats, approximately 200 genes could be involved in this network in the *P. taeda* genome. In *P. lambertiana*, insertions of *Irbe TE* within 143 high-quality gene introns were found, many of which are important genes coding for transcription factors, chromatin modification enzymes, protein kinases, and receptors. The short *Plater* MITE family identified in proximal gene-flanking regions and introns could provide TATA boxes and several ARR1, DOF, W-box, GT, and MYB-binding sites, which are important signals in plant transcription activation and stress-response regulation [111,112,113]. Ten different ARR1 binding sites (seven on the positive and three on the negative strand) were present in the consensus sequence of the *P. taeda Plater* element. The transcription factor-type response regulator ARR1 directs transcriptional activation of the ARR6 gene, which responds to cytokinins without *de novo* protein synthesis [114]. Cytokinins are an important class of phytohormones that are involved in developmental processes and growth [115,116,117], as well as in defense responses [118]. The (AG)_4_A motif is one of the most common TFBS for plant promoters [119] that regulate light-responsive phototransduction processes in plants [120]. The LTR RLX *Daugava* identified in this study contains longer AG-rich tracts; this RLX family is distributed in pine gene introns and several flanks and might form an important responsive gene network. Genome-wide approaches in mammalian genomes demonstrate that TEs contribute to rewiring and selection of gene networks [121,122]. Approximately one-sixth of rice genes are associated with TEs [123]. Comparative analysis of three inbred maize lines revealed that the expression of 33% of stress-responsive genes could be attributed to regulation by TEs [124]. Various maize TEs contain approximately 25% of all DNase I hypersensitivity sites within the genome, which are associated with open chromatin and cis-acting elements and are therefore essential transcriptional regulators [125]. In contrast to mammals, plant genes more commonly contain longer introns, which are expressed at higher levels [126,127]. However, in recent studies, expression in different tissues or expression breadth has also been considered, and plant genes expressed across a wide range of tissues or conditions were found to have a higher intron density [128,129]. In *P. taeda* and *P. lambertiana*, extensive transcriptome data are not currently available; therefore, the role of TE insertions within introns in regulation of gene expression requires additional investigations. Using the *Daugava* RLX gene network identified in this study, the formation of specific patterns or intronic genotypes was tested. TE patterns could influence gene availability, responsiveness, stability, or higher order organization structures in the nucleus. Several revealed genes with unique *Daugava* RLX insertions are involved in the process of pre-mRNA maturation and splicing, while other genes with identical combinations of TE insertions are linked to protein metabolic processes and Golgi body homeostasis, which are related processes. Further investigation will enable more thorough analyses of relevance. The function of genes where multiple TEs were identified within introns (e.g., potassium channel coding genes and other receptors, protein kinases, cytochrome genes) suggests involvement in the maintenance of cell homeostasis under stress conditions. These genes were found to have many co-occurring GO terms, indicating that gene products are involved in many cellular processes and these genes may be expressed or retain stability under a broad range of conditions. We suggest that genes with many different types of TEs could act as node genes that are functional or stable across a range of conditions and could be important in early defense responses and rapid metabolome switching. This is also supported by the discovery of several homologous genes with large introns in both pine species. Cases of independent transpositions of different RLX families into homologous genes in varieties of differing origin resulting in similar phenotypes have been reported [22]. Additionally, the TE insertion patterns in the investigated pine introns were found to have a lower average GC content (39%) than nearby transcripts. The GC content of gene transcripts in the studied genes in *P. taeda* and *P. lambertiana* was comparable (44%) and higher than the reported genome average of 38% [91,92]. Similarly, a lower GC content in plant gene introns has been reported for other plant species [130,131], indicating that intron sequences may have a more relaxed DNA conformation and are more accessible to transcription and other regulatory factors [132,133].

It remains unclear if TE sequences have insertional preferences, but it has been reported that a relaxed chromatin configuration promotes TE insertions and leads to an increased mutation probability in tissue- and stage-specific genes [134,135]. In this aspect, TEs should be randomly distributed in different stress-responsive gene non-coding regions (flanks and introns). The non-autonomous *Plater* MITE element insertions were statistically significantly overrepresented in the proximity of pine genes (0–2 kb), a distance over which linkage equilibrium extends in conifers [136]. MITEs are preferentially located within gene regions of many plants and could influence gene expression [46,47,137,138,139]. There is evidence that some MITE insertions located close to gene promoters could also downregulate gene expression [140]. *Plater* was also frequently inserted in the introns of genes of the studied pine species. However, in this study, no genes with several *Plater* insertions in different non-coding regions (e.g., flanks and introns) were identified. This distribution suggests the formation of differentially regulated gene sub-networks, depending on the location of MITE insertions. For example, it was determined that many genes containing *Plater* in flanking regions are involved in the regulation of developmental processes and cell division, but genes having *Plater* in their introns are associated with immune responses and cell-wall biosynthesis, among other activities. Therefore, we could hypothesize about the mechanisms of action of gene-TE associated networks. If *Plater* cis-acting elements in the more accessible introns help to activate a specific network of stress-responsive genes, then non-coding RNA products of splicing from these genes could be involved in feedback loop mechanisms for blocking transcription of genes with flanking *Plater* insertions, near genes involved in developmental processes. In this example, identical TEs initiate transcription of defense genes under unfavorable conditions, while at the same time have a down-regulatory effect on developmental gene transcription, switching to an energy-saving mode. Similar strategies could be highly advantageous for the rapid activation of defense responses and switching of metabolic functions. However, it is only one explanation among the other possibilities and this could be a subject of further studies. Additionally, in the current study, *Plater* MITE insertions were found in both analyzed pine species, belonging to separate subgenera, suggesting similar distributions also in other pine species. Therefore, *Plater* could be a useful molecular marker for genotyping of pine species, as shown for MITEs in other plant species [48,141,142]. 

TE distribution is linked with speciation [143,144]. In pines, transposition activity accounts for the period following the divergence of the pine subgenera *Strobus* (*P. lambertiana*) and *Pinus* (*P. taeda*) and speciation [62,65,67,68,145]. Only a few homologous genes in both pine species were found with similar TE insertions, indicating that differences in TE family distribution between species are also found in more conservative gene regions. Insertion of the *Irbe* DNA TE in homologous genes of both species was found in PSMD4, a 26S proteasome non-ATPase regulatory subunit gene (40% query coverage, 98% sequence identity of transcripts). Insertion of the ancient LTR RLX *IFG* [67] was found in three homologous *P. taeda* and *P. lambertiana* gene pairs, surprisingly, all encoding different protein kinases. However, some of the *IFG* insertions were located in different introns of kinase genes and therefore could also represent independent transpositions. Analyzed *IFG* sequences contained masked regions and therefore the age of the insertions was not evaluated. Notably, insertions of other TEs were also frequently found in different receptor-like protein kinases and tyrosine protein kinases in our study. Protein kinase genes belong to one of the most proliferated gene superfamilies in plants, whose members are linked with a range of key metabolic and various plant-specific adaptation processes [146,147,148,149].

In conclusion, the source of TE sequences expressed in response to stress conditions could be the transcription of introns of many stress-responsive genes, which could explain the observed highly correlated expression levels of RLX families within individuals [70]. Transfer of information about TE insertions in gene regions to non-model pine species is complicated, as common TE families were revealed, but they are generally located in non-homologous genes. This highlights the need for additional studies and sequencing of species of interest to investigate TE-associated polymorphisms, such as in *P. sylvestris*, which is an important species in northern Europe. In the two analyzed pine species, TE insertions were more often found in gene introns but less commonly in gene-flanking regions, similar to other plant species [150,151,152,153]. Insertions of TEs in gene regions are associated with various epigenetic mechanisms, but it remains unknown how some actively transcribed plant genes cope with large introns [140]. Cultivation and plant breeding reduce genetic diversity and fitness to environmental stresses [154]. A recent whole-genome comparison of wild and cultivated rice species revealed depletion of intronic TE insertions in cultivated species [155]. Gymnosperms are outcrossing species that produce large quantities of pollen and seeds, generating a genetically diverse germplasm pool for subsequent natural selection of highly adaptable seedlings. Pine species are known for their strong adaptation to local growing conditions [156,157,158,159]. This study demonstrated the increased accumulation of TE sequences in stress-responsive gene introns, and suggested the rewiring of them into responsive networks interconnected with node genes containing multiple TEs. However, this hypothesis requires further investigations. Preferential TE insertion in open chromatin has been reported for tissue-specific genes [134], hence similar mechanisms followed by sustained natural selection could drive the accumulation of adaptive TEs in gene non-coding sequences of plants. The inclusion or exclusion of genes from TE-mediated networks is an efficient means of effecting dynamic changes in response to various environmental factors, including changing host–pathogen interactions and multifactorial processes in plants such as communication and signaling organization. Many such regulatory influences could lead to the adaptive environmental response clines that are characteristic of widely distributed pine populations [160,161,162,163].

## Figures and Tables

**Figure 1 genes-11-01216-f001:**
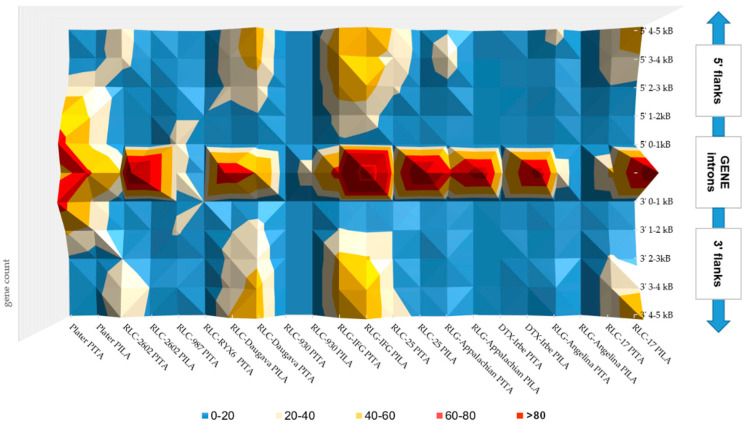
Comparison of TE distribution in gene non-coding regions of high-quality genes of the *P. lambertiana* genome v.1.01 (PILA) and filtered annotated gene set of *P. taeda* v.2.0 (PITA).

**Figure 2 genes-11-01216-f002:**
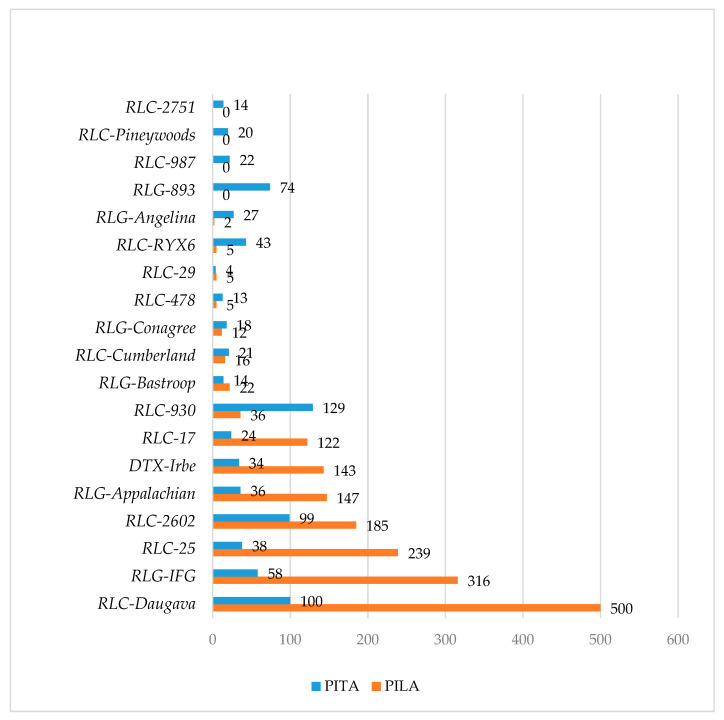
Distribution of >1 kb hits to TEs in gene introns of filtered *P. taeda* (PITA) genes and high-quality *P. lambertiana* (PILA) genes.

**Figure 3 genes-11-01216-f003:**
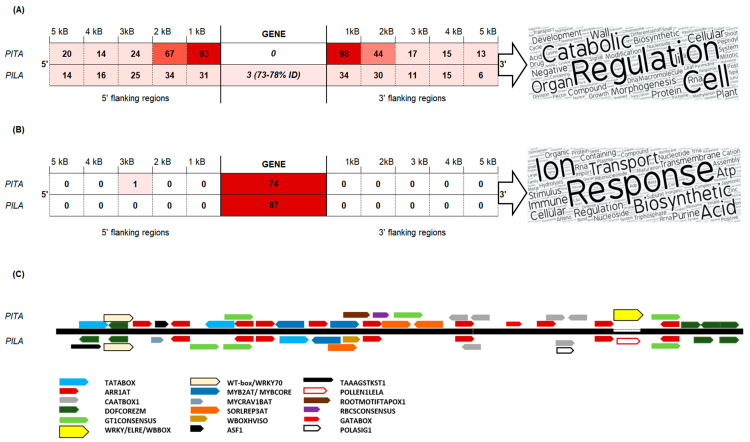
(**A**) Distribution of *Plater MITE* insertions across *P. taeda* (PITA) and *P. lambertiana* (PILA) gene-flanking regions. (**B**) Distribution of *Plater MITE* insertions across *P. taeda* (PITA) and *P. lambertiana* (PILA) gene introns. World cloud generated from biological process GO terms of *P. taeda* genes involved in the networks using the online tool https://wordart.com/. (**C**) Alignment of *P. taeda* (PITA) and *P. lambertiana* (PILA) consensus sequences with predicted plant cis-acting regulatory elements.

**Figure 4 genes-11-01216-f004:**
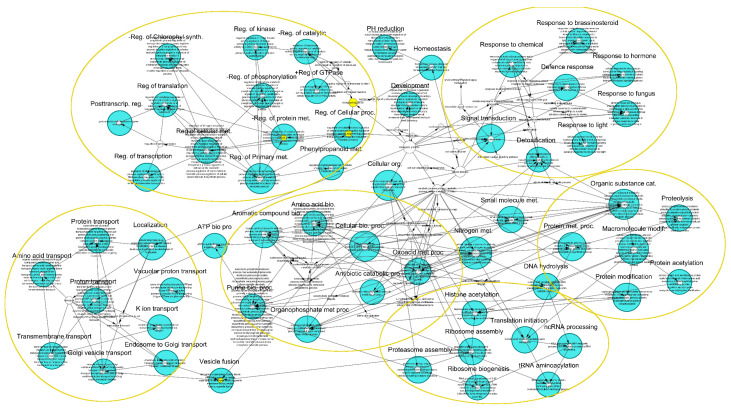
Gene network formed by *Irbe* DNA TE presence in the gene introns of *P. taeda* v.2.0.

**Table 1 genes-11-01216-t001:** Comparison of extracted gene-flanking regions in genome data sets.

Genome/Gene Set	Flanking Region from the Gene Start/End Coordinates
5′	3′	5′	3′	5′	3′	5′	3′	5′	3′
0–1 kb	0–1 kb	1–2 kb	1–2 kb	2–3 kb	2–3 kb	3–4 kb	3–4 kb	4–5 kb	4–5 kb
***P. taeda v.2.0.*** ***all genes***	Nb of extr.seq.	36,726	36,728	34,711	34,063	33,184	32,310	31,767	30,838	30,349	29,479
Nb of hqh to TE-dr	5851	6450	4362	3901	3750	3628	3310	3069	3202	2924
ratio	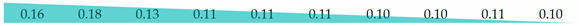
>50	17	22	10	10	4	2	1	0	0	0
>100	8	9	1	0	0	0	0	0	0	0
***P. taeda v.2.0.*** ***annotated genes***	Nb of extr.seq.	15,084	15,057	14,114	13,793	13,371	12,912	12,713	12,192	11,985	11,569
Nb of hqh to TE-dr	816	773	800	732	875	968	1161	991	901	1000
ratio	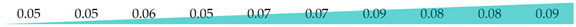
>50	0	0	0	0	0	0	0	0	0	0
>100	0	0	0	0	0	0	0	0	0	0
***P. taeda v.1.01.***HQ genes	Nb of extr.seq.	4298	4239	4177	4128	4130	4091	4081	4028	4023	3967
Nb of hqh to TE-dr	784	779	2258	1890	3151	2693	3593	3222	3816	3539
ratio	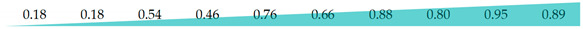
>50	1	1	1	0	0	0	0	0	0	0
>100	0	0	0	0	0	0	0	0	0	0
***P. taeda v.1.01.***LQ genes	Nb of extr.seq.	75,425	75,459	72,840	72,797	71,554	71,470	70,002	69,836	68,237	68,017
Nb of hqh to TE-dr	2317	2540	4188	4243	4979	5070	5256	5387	5645	5382
ratio	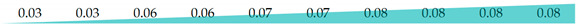
>50	2	2	5	5	6	5	4	7	7	6
>100	1	1	3	4	1	1	0	1	0	0
***P. lambertiana v.1.01***HQ genes	Nb of extr.seq.	8779	8778	8746	8742	8719	8708	8692	8673	8660	8640
Nb of hqh to TE-dr	71	55	163	187	278	277	315	296	355	357
ratio	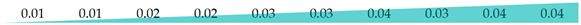
>50	0	0	0	0	0	0	0	0	0	0
>100	0	0	0	0	0	0	0	0	0	0
***P. lambertiana v.1.01*** ***LQ***	Nb of extr.seq.	71,162	71,157	70,386	70,475	69,773	69,909	69,217	69,344	68,660	68,836
Nb of hqh to TE-dr	470	466	1063	1011	1556	1508	1789	1368	2038	1999
ratio	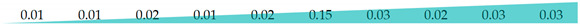
>50	0	0	1	0	4	3	6	1	7	7
>100	0	0	0	0	0	0	0	0	0	0

>—indicates count of TE families that hits greater than 50 or greater than 100 gene flanking regions; hqh—high quality hits; TE-dr—TE-derived repeats; HQ—high quality gene set; LQ—low quality gene set; extr.seq.—extracted gene flanking sequences.

**Table 2 genes-11-01216-t002:** *P. taeda* v.2.0 and *P. lambertiana* v.1.01 genes carrying several *Plater* MITE insertions. qc- query coverage.

Species	Genes ID with Multiple *Plater MITEs*	Nb of *Plater* Insertions	Description	qc, %	*e*-Value	ID, %	Accession
*P. taeda* v.2.0.	PITA_12742	7	uncharacterized protein with domain of phosphoglucosamine mutase family protein	88	0.00 × 10^0^	65	PLN02371
	PITA_21987	4	subtilisin-like protease SBT5.3	96	0.0	47	XP_012083905.1
	PITA_00114	3	metal tolerance protein 11	99	0.0	72	XP_006857671.1
	PITA_24114	2	probable xyloglucan endotransglucosylase/hydrolase protein B	93	3.00 × 10^−153^	72	XP_030961064.1
	PITA_21327	2	60S ribosomal protein L8-1-like	95	2.00 × 10^−169^	90	XP_022936671.1
	PITA_17959	2	TMV resistance protein N-like	93	8.00 × 10^−165^	31	XP_023886681.1
	PITA_34859	2	3-oxoacyl-[acyl-carrier-protein] synthase I, chloroplastic-like isoform X1	95	0.0	74	XP_028101593.1
	PITA_28894	2	L-gulonolactone oxidase 2 isoform X2	95	0.0	54	XP_011621860.1
	PITA_00539	2	probable potassium transporter 11	99	0.0	67	XP_006830082.1
	PITA_33316	2	plasma membrane intrinsic protein 2;8	95	5.00 × 10^−141^	74	NP_179277.1
	PITA_09881	2	cytokinin hydroxylase	93	0.0	52	XP_011099558.1
*P. lambertiana* v.1.01. HQ genes	S/hiseq/c38458_g1_i1|m.23006	2	bifunctional phosphatase IMPL2, chloroplastic	75	7.00 × 10^−147^	73	XP_011088446.1
	PILAhq_048992	2	putative clathrin assembly protein At4g40080	80	2.00 × 10^−40^	36	XP_027337607.1
	PILAhm_002002	2	histone deacetylase 15 isoform X3	69	4.00 × 10^−179^	63.89	XP_010265267.1

HQ—high quality genes; qc—query coverage; ID—identity; Nb—number.

**Table 3 genes-11-01216-t003:** Node genes containing several TE insertions and found to be homologous or carrying identical domains between *P. taeda* (PITA) and *P. lambertiana* (PILA).

TE-dr Nb. *Pita*	TE-dr Nb. *Pila*	Description	Accession	Conserved Domain Name	Accession	GO Terms
24	19	plastidial pyruvate kinase 2	XP_006843356.1 ^h^	PLN02623	PLN02623	reproduction; ATP generation from ADP; seed maturation;
23	26	DEAD-box ATP-dependent RNA helicase 20 isoform X2/helicase 58, chloroplastic isoform X3	XP_025888827.1	SrmB	COG0513	RNA secondary structure unwinding
21	21	phospholipid:diacylglycerol acyltransferase 1	XP_006849611.1 ^h^	PLN02517	PLN02517	acylglycerol biosynthetic process
18	20	nuclear pore complex protein NUP62-like/GPCR-type G protein 1 isoform X2	XP_024396806.1	SMC_prok_B super family	cl37069	RNA export from nucleus; protein import/export into/from nucleus; nucleocytoplasmic transport, localization
13	23	WD repeat-containing protein WRAP73	XP_008798782.1	WD40 super family	COG2319	-
-	19	protein RAE1	XP_028076289.1		cl29593	
-	24	actin-related protein 2/3 complex subunit 1A	XP_011627051.1		cl29593	
12	19	uncharacterized protein LOC109715170/probable E3 ubiquitin-protein ligase HERC4 isoform X1	XP_020095639.1	ATS1 super family	cl34932	-
11	31	peroxisomal adenine nucleotide carrier 1/mitochondrial substrate carrier family protein C-like	XP_006841423.1	Mito_carr	pfam00153	Establishment of localization; transmembrane transport; amide biosynthetic process; translation; nitrogen compound metabolic process.

TE-dr Nb—Number of unique TE-derived repeats; ^h^—genes with similar transcripts (>90% nucleotide identity).

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
