# Peer review of "Comparative Study of Pine Reference Genomes Reveals Transposable Element Interconnected Gene Networks"

_genes, 2020, doi:10.3390/genes11101216_

Round 1
Reviewer 1 Report
The manuscript by Voronova et al. described bioinformatics analysis done in two pine species. They address the complexity of transposable elements in conifer and the difficulty in resolving these issues in such large genomes. The advance in long reads sequencing technologies is now making it possible. The authors found a possible correlation in the increase of TEs in introns to the adaptative environmental response that could explain their high environmental tolerance, although there is no evidence to support this. The findings confirm the need to re-sequence the available genomes for conifers that are available using short-read assemblies that frequently leads to errors in the genes coordinates and TEs annotation due to technical limitations. Only a few gymnosperms species have been sequenced and assembled to date, limiting proper annotation and further conclusions regarding biological relevance for those species. The study is relevant for the scientific community shedding light on the importance of conifers TEs in transcript modulation, but the data are too limited to make sweeping conclusions.
I have no major concerns.
Minor issue.
The discussion in this paragraph is speculative, as there is no data presented to support it. It should be made clear that this is "one speculative scenario among other possibilities".
- MITE3321 was frequently inserted also in
- 611 the introns of genes of the studied pine species. However, in this study, no genes with several
- 612 MITE3321 insertions in its different non-coding regions (flanks or introns) were identified. This
- 613 distribution suggests formation of differentially regulated gene sub-networks, depending on the
- 614 location of MITE insertions. For example, it was determined that many genes containing MITE3321
- 615 in flanking regions were involved in the regulation of developmental processes and cell division, but
- 616 genes having MITE3321 in their introns were associated with immune responses and cell-wall
- 617 biosynthesis, among other activities. If MITE3321 cis-acting elements in the more accessible introns
- 618 help activate a specific network of genes, then non-coding RNA products of splicing from stress-
- 619 responsive genes could be involved in feedback loop mechanisms for blocking transcription of genes
- 620 with proximal MITE3321. In this example, identical TEs could have a downregulatory effect on
- 621 developmental gene transcription, switching to an energy-saving mode, while at the same time
- 622 increasing transcription of defense genes. Similar strategies could be highly advantageous for the
- 623 rapid activation of defense responses and switching of metabolic functions. Additionally, in the
- 624 current study, MITE3321 insertions were found in both analyzed pine species, belonging to separate
- 625 subgenera, suggesting similar distributions also in other pine species. Therefore, MITE3321 could be
- 626 a useful molecular marker for genotyping of pine species, as shown for MITEs in other plant species
- 627 [48, 140, 141].
Author Response
Thank you for your review and suggestions. The paragraph in question is in the discussion and it contains our hypothesis. In a revised version of the manuscript, we add a sentence that explains this, to make it more clear for the readers.
Of course, further studies on gene function, expression, MITE3321 insertional polymorphism in the populations of each species, etc are needed for the assessment of hypothesis approval or denial.
Reviewer 2 Report
Review of manuscript entitled “Comparative study of pine reference genomes reveals transposable element interconnected gene networks » by Voronova et al.
”
In this study, Veronova et al. analyze transposable elements from two pine species: Pinus taeda and Pinus Lambertiana.
Characterizing Transposable Element content, variation among species and impact on gene expression is important to better understand the molecular bases of genome evolution. This is thus a topic of interest, and it fits well with the “Transposable elements in plant genomes” special issue of Genes.
While the topic is interesting, I have hard time understanding what question the authors have addressed. The manuscript would be largely improved if the authors were better presenting the rationale of each analysis, followed by the corresponding results. The text is globally difficult to read, which makes it difficult to understand whether the results are sound. The authors provide some interesting information on TE localization and annotation, but comparison of the two genomes is difficult to follow. Most results are clear tendencies, but statistical tests are lacking. Some results are not supported by any table or figure. The methodology used for network analysis is not clear enough, which does not allow to be sure that the results are sound. The Discussion is too long, a more focused redaction would improve it. Figures and Tables lack some legend information.
Author Response
Thank you for your review and suggestions.
- I have hard time understanding what question the authors have addressed. The manuscript would be largely improved if the authors were better presenting the rationale of each analysis, followed by the corresponding results. The text is globally difficult to read, which makes it difficult to understand whether the results are sound.
The manuscript has been carefully examined and revised to improve the understanding of the text. We have stated clearly our aims in the article: at the end of the introduction: lines 87-93.
Results contains sections with headings, and we constantly provide reasoning and explain the flow of the experiments: line 139-140; line 154-155; line 176-179; line 207-211; line 243-245; line 311-316; line 341-342; line 374-375; line 396-401; line 446-448. As well, in the method section lines 758-759, we have provided a link to a graphical overview of analysis workflow (Additional file 1).
- The authors provide some interesting information on TE localization and annotation, but comparison of the two genomes is difficult to follow. Most results are clear tendencies, but statistical tests are lacking. Some results are not supported by any table or figure.
Only interspersed repeats distributed in gene regions of the two pine reference genomes were compared, but certain TE elements with distribution in both genomes were studied in the details. Only one genotype genomic sequence per pine species is currently available. Of course, these TE insertions could vary in the populations of each species, but this information is unavailable currently and it is aimed for the future studies.
All described results are supported with data, however, most data are in the Supplemental material (available currently in the supplementary section of pre-print (https://www.researchsquare.com/article/rs-34803/v1), as excel files with multiple tabs were not supported for the upload to the MDPI system. Due to the large size of tables, not everything could be included in the article even in the Supplementary section, not all data could be illustrated. As authors, we are responsible for all information included. Please indicate, which currently unsupported results should be supported, as we would like to be able to improve the understanding of the readers if such a problem exists.
Appropriate statistical tests and quality control were performed for all results obtained. For example, TE topology in the gene flanks was accessed by the t-test of TE presence comparison between regions depending on the distance from the protein-coding gene (line176-186; 708-712), in Additional file 4 only significantly overrepresented repeats in the vicinity from genes were presented. For the evaluation of gene networks, data quality assessment was performed in the stage of identification of high-quality sequence matches in filtered protein-coding genes (for the final networks, we considered matches longer than 1kb, with higher than 80% sequence identity (e-values <0,001)). Default quality frames are integrated into all bioinformatics tools we have used for the analyses performed (see method section for the reference).
Please, indicate exactly, which tests are lacking otherwise it is hard to discuss an issue further.
- The methodology used for network analysis is not clear enough, which does not allow to be sure that the results are sound.
Thank you for this suggestion, we have updated this section in the new version of the article. Each gene network was established by the presence of a high-quality match to TE (members of one TE family), present in a similar location of genomic sequence (0-1 kb gene flanks or introns). For the evaluation of gene networks, quality control was assessed on the stage of identification of high-quality sequence matches (>1kb, >80% sequence identity to reference sequence of genes; for shorter elements like MITEs, hit longer than 200 bp was considered). Therefore, only genes containing hit to TE were evaluated, and a group of such genes was used then for the networking analysis. If some members of such networks were annotated, overall functional aspects of this suggested network were accessed. Therefore, gene networks were united by the presence of the insertion of one TE family member, which contains certain TFBS or other signals.
For the update of the gene annotations, transcripts from associated gene networks (only for those significant networks, included in the publication) were additionally annotated with Blast2GO tool (CLC Genomic workbench), which uses cloud-BLAST to NCBI reference protein database, Additional file 7.
- The Discussion is too long; a more focused redaction would improve it.
We have improved some issues in the discussion. We would like to discuss all the aspects evaluated in this study, as some of them could be important for other researchers or for future studies.
Discussion is split in logical sections, according to the described results (Reference genome quality assessment; TE quality assessment; Evaluated important TEs and associated regulatory motifs; suggested gene networks and hypothetical mechanisms; homologous genes; main hypothesis &conclusions).
- Figures and Tables lack some legend information.
We have noticed errors in Table 1 and corrected them in the newer version of the manuscript. Please indicate, if Table 1 was an issue. If the information is lacking elsewhere, please do not hesitate to indicate exactly, so that we could correct the mistakes.